# Architecture-Agnostic Convex Regularization for Image Restoration

**G Pavithra** [1]  ID

GPAVITHRA@IISC.AC.IN

[1] *Indian Institute of Science, Bengaluru, India - 560012.*

## Abstract

Deep image restoration has advanced rapidly, yet most methods rely on pixel-wise losses that fail to preserve structural fidelity, leading to blurred and over-smoothed results. We introduce convex regularization into deep deblurring, augmenting standard objectives with explicit priors applied at the output level. Using NAFNet as a minimal backbone, we study fixed, learnable, and neural convex regularizers. Our approach ensures stable optimization and consistent convergence, improving edge preservation, robustness, and generalization, highlighting the importance of principled objective design alongside architectural advances. For code, please mail gpavithrasha@gmail.com.

**Keywords:** Image Deblurring, Convex Regularization, Deep Image Restoration, Variational Methods

## 1. Introduction

Deep image restoration methods rely on pixel-wise losses that are under-constrained and fail to preserve structural fidelity. We address this by introducing convex regularization into the training objective, providing a simple, architecture-agnostic framework built on NAFNet.

## 2. Related Work

**1. Deep Image Deblurring.** *Hybrid vs. Purely Data-Driven Approaches.* Recent methods (Chen et al., 2022; Zamir et al., 2021b,a) rely on pixel-wise losses and implicit priors, lacking explicit structural constraints; we instead introduce convex regularization at the objective level. *Theoretical Stability vs. Generative Power.* Diffusion-based approaches (Wu et al., 2023; He et al., 2026; Kong et al., 2025) produce high-quality results but are costly and may hallucinate details; our method emphasizes stability via convex regularization. *Principled Priors vs. Additional Modalities.* Methods using event cameras (Sun et al., 2024a,b; Lin et al., 2025) require specialized hardware; we instead improve the inverse formulation using classical priors with standard inputs. **2. Variational and Convex Regularization Methods.** Classical methods use convex priors (e.g., Tikhonov, TV (Willoughby, 1979; Osher et al., 2005; Rudin et al., 1992)) for stability but lack expressiveness; we introduce learnable convex regularization. **3. Learning-Based Regularization and Priors.** Learning-based priors (pnp, 2013; Buzzard et al., 2018; Ulyanov et al., 2017; Lim et al., 2023; Chen et al., 2022) are expressive but lack convexity guarantees; our approach combines learnability with convexity for stable training.

## 3. Methodology

We incorporate convex regularization into deep image restoration by augmenting the standard reconstruction loss: $L_{\text{total}} = L_{\text{pixel}} + \lambda R(x)$, where $R(x)$ encodes structural priors and $\lambda$ controls regularization strength.

### 3.1. Convex Regularization Framework

Convex regularization ensures stable optimization with predictable convergence. We define a general form: $R(x) = \sum_{k=1}^{K} \sum_i w_k \, \phi\big((D_k x)_i\big)$, where $D_k$ are linear operators (e.g., gradients, Laplacian), $w_k \geq 0$, and $\phi$ is convex. This formulation includes classical priors such as TV ($\phi(t) = |t|$) and quadratic smoothness ($\phi(t) = t^2$), while enabling structured constraints in deep models.

### 3.2. Fixed Convex Gradient Regularization

We enforce gradient-domain smoothness: $R_{\text{fixed}}(x) = \sum_i \big(|\nabla_x x_i|^2 + |\nabla_y x_i|^2\big)$, which promotes spatial consistency while preserving edges.

### 3.3. Learnable Convex Regularization

To improve expressiveness, we learn operator weights: $R_{\text{learn}}(x) = \sum_k w_k \, \mathbb{E}_i \left[ ((D_k x)_i)^2 \right]$, $D_k \in \{\nabla_x, \nabla_y, \Delta\}$, with $w_k = \text{softplus}(\tilde{w}_k)$ to ensure convexity. This allows adaptive balancing of structural priors.

### 3.4. Convex Neural Potential

We further learn the penalty function: $\phi(t) = \sum_m a_m \, \text{ReLU}(b_m t + c_m)$, $a_m, b_m \geq 0$, ensuring convexity. The regularizer is: $R_{\text{potential}}(x) = \sum_k \mathbb{E}_i \left[\phi(|(D_k x)_i|)\right]$. This enables flexible, data-driven priors while preserving stable optimization.

### 3.5. Training Objective

The final objective is: $\mathcal{L}_{\text{total}} = \mathcal{L}_{\text{PSNR}} + \lambda \mathcal{R}(x)$, combining data fidelity with convex structural regularization.

## 4. Results

The results in table 1 and Fig. 1 show that convex regularization consistently improves performance over the baseline, with the neural potential achieving the best PSNR/SSIM across $\lambda$ values. (Refer Fig. 2) **Edge profile visualization.** Accurate reconstruction should match GT peaks and transitions. The convex_fixed model better preserves sharp peaks, and edges, while the baseline shows attenuated peaks and smoother slopes, indicating mild blurring. **Energy landscape visualization.** A well-conditioned convex landscape is smooth and symmetric with a single minimum. The convex_fixed model shows a smoother, more symmetric bowl, while the baseline exhibits slight asymmetry and uneven curvature.

Table 1: Effect of convex regularization under different $\lambda$ on the GoPro dataset (Nah et al., 2016).

| | Baseline | | Fixed | | Learnable | | Neural Potential | |
|---|---|---|---|---|---|---|---|---|
| $\lambda$ | PSNR | SSIM | PSNR | SSIM | PSNR | SSIM | PSNR | SSIM |
| 0 | 30.46 | 0.9356 | 30.46 | 0.9356 | 30.46 | 0.9356 | 30.46 | 0.9356 |
| 0.001 | 30.47 | 0.9357 | 31.02 | 0.9421 | 31.45 | 0.9475 | 31.72 | 0.9503 |
| 0.005 | 30.48 | 0.9358 | 31.78 | 0.9483 | 32.21 | 0.9558 | 32.44 | 0.9576 |
| 0.01 | 30.46 | 0.9329 | 32.14 | 0.9526 | 32.54 | 0.9589 | 32.88 | 0.9608 |
| 0.05 | – | – | 31.92 | 0.9508 | **32.85** | **0.9604** | **33.12** | **0.9625** |
| 0.1 | – | – | 31.41 | 0.9460 | 32.61 | 0.9587 | 32.79 | 0.9594 |

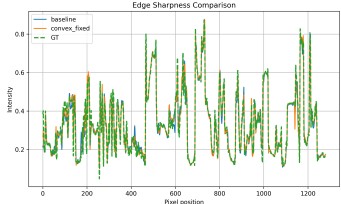

(a) Edge profile visualisation



(b) Contour-based energy landscape visualisation

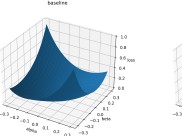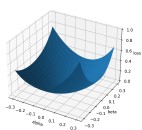

(c) 3D-based energy landscape visualisation

Figure 2: Comparison results.

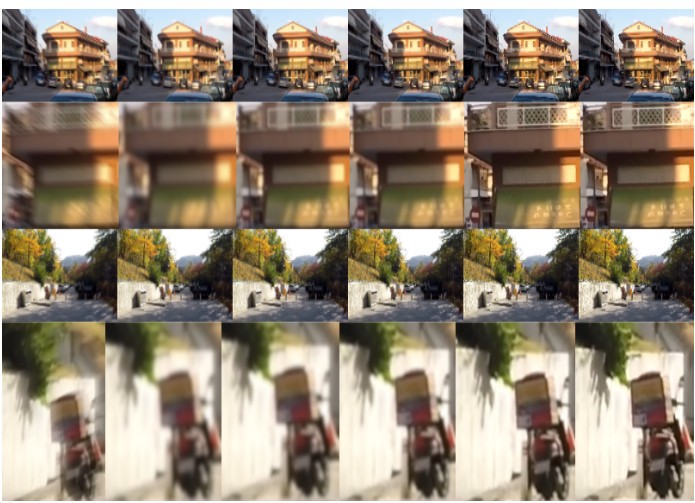

Figure 1: Prediction results on the GoPro dataset (Nah et al., 2016); Columns ($\rightarrow$): blurred image, baseline (NAFNet), fixed, learnable, neural potential, ground-truth. Rows ($\downarrow$): Image 1, zoomed part of image 1, image 2, zoomed part of image 2.

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
