# OpenReview forum: "Architecture-Agnostic Convex Regularization for Image Restoration"
_MIDL.io/2026/Short_Papers — MIDL 2026 - Short Papers Poster_

### Official Review · Reviewer_kmVm · 2026-05-04
**Convex regularization for deep image restoration**

**Rating:** 4
**Confidence:** 5

**Review:**

Overall, the paper presents a clean and intuitive extension of classical convex regularization into modern deep learning-based restoration frameworks. The formulation is straightforward and well-grounded in variational principles.

Strengths:
- Simple and general framework that can be applied across architectures
- Clear connection to classical variational methods (e.g., TV, quadratic priors)
- Consistent improvements in quantitative metrics
- Well-motivated emphasis on objective design and stability

Weaknesses:
- Evaluation is restricted (single dataset, single backbone)

Overall, despite moderate novelty, the method is well-executed and practically relevant, and the results support the effectiveness of the approach.

**Summary:**

This paper introduces an architecture-agnostic framework for incorporating convex regularization into deep image restoration, built on top of a NAFNet backbone. The method augments standard pixel-wise losses with convex priors, including fixed, learnable, and neural potential-based regularizers, while maintaining convexity constraints. Experiments on the GoPro dataset show consistent improvements in PSNR and SSIM, with the neural potential variant achieving the best performance. The work aims to highlight the importance of principled objective design for improving structural fidelity and optimization stability.

**Strengths:**

The paper presents a clear and principled framework for integrating convex regularization into deep image restoration. The method is simple, modular, and architecture-agnostic, making it broadly applicable. The connection to classical variational methods is well established, and the extension to learnable and neural convex potentials is a natural progression. Experimental results show consistent improvements over the baseline, and the formulation promotes stable optimization and better structural preservation.

**Weaknesses:**

The experimental evaluation is somewhat limited, as it is conducted on a single dataset and backbone. While the improvements are consistent, broader validation across different datasets and architectures would further strengthen the generality and impact of the conclusions.

**Justification Of Rating:**

The paper offers a clean and practically useful contribution with experimental results. While the novelty is moderate and evaluation scope is limited, the method is well-motivated, easy to adopt, and likely beneficial to the community.

---

### Decision · Program_Chairs · 2026-05-08

Accept (Poster)